# Adversarial Robustness of Streaming Algorithms through Importance Sampling

**Vladimir Braverman**
Google
vbraverman@google.com

**Avinatan Hassidim**
Google
avinatan@google.com

**Yossi Matias**
Google
yossi@google.com

**Mariano Schain**
Google
marianos@google.com

**Sandeep Silwal**
MIT
silwal@mit.edu

**Samson Zhou**
Carnegie Mellon University
samsonzhou@gmail.com

## Abstract

Robustness against adversarial attacks has recently been at the forefront of algorithmic design for machine learning tasks. In the adversarial streaming model, an adversary gives an algorithm a sequence of adaptively chosen updates $u_1, \ldots, u_n$ as a data stream. The goal of the algorithm is to compute or approximate some predetermined function for every prefix of the adversarial stream, but the adversary may generate future updates based on previous outputs of the algorithm. In particular, the adversary may gradually learn the random bits internally used by an algorithm to manipulate dependencies in the input. This is especially problematic as many important problems in the streaming model require randomized algorithms, as they are known to not admit any deterministic algorithms that use sublinear space. In this paper, we introduce adversarially robust streaming algorithms for central machine learning and algorithmic tasks, such as regression and clustering, as well as their more general counterparts, subspace embedding, low-rank approximation, and coreset construction. For regression and other numerical linear algebra related tasks, we consider the row arrival streaming model. Our results are based on a simple, but powerful, observation that many importance sampling-based algorithms give rise to adversarial robustness which is in contrast to sketching based algorithms, which are very prevalent in the streaming literature but suffer from adversarial attacks. In addition, we show that the well-known merge and reduce paradigm in streaming is adversarially robust. Since the merge and reduce paradigm allows coreset constructions in the streaming setting, we thus obtain robust algorithms for $k$-means, $k$-median, $k$-center, Bregman clustering, projective clustering, principal component analysis (PCA) and non-negative matrix factorization. To the best of our knowledge, these are the first adversarially robust results for these problems yet require no new algorithmic implementations. Finally, we empirically confirm the robustness of our algorithms on various adversarial attacks and demonstrate that by contrast, some common existing algorithms are not robust.

## 1 Introduction

Robustness against adversarial attacks have recently been at the forefront of algorithmic design for machine learning tasks [GSS15, CW17, AEIK18, MMS$^+$18, TSE$^+$19]. We extend this line of work by studying adversarially robust streaming algorithms.

In the streaming model, data points are generated one at a time in a stream and the goal is to compute some meaningful function of the input points while using a limited amount of memory, typically

35th Conference on Neural Information Processing Systems (NeurIPS 2021).

*sublinear* in the total size of the input. The streaming model is applicable in many algorithmic and ML related tasks where the size of the data far exceeds the available storage. Applications of the streaming model include monitoring IP traffic flow, analyzing web search queries [LMV+16], processing large scientific data, feature selection in machine learning [HZZ21, GRB+19, WYWD10], and estimating word statistics in natural language processing [GDC12] to name a few. Streaming algorithms have also been implemented in popular data processing libraries such as Apache Spark which have implementations for streaming tasks such as clustering and linear regression [ZXW+16a].

In the adversarial streaming model [MBN+17, BMSC17, AMYZ19, BY20, BJWY20, HKM+20, WZ20, ABD+21, KMNS21], an adversary gives an algorithm a sequence of adaptively chosen updates $u_1, \ldots, u_n$ as a data stream. The goal of the algorithm is to compute or approximate some predetermined function for every prefix of the adversarial stream, but the adversary may generate future updates based on previous outputs of the algorithm. In particular, the adversary may gradually learn the random bits internally used by an algorithm to manipulate dependencies in the input. This is especially problematic as many important problems in the streaming model require randomized algorithms, as they are known to not admit any deterministic algorithms that use sublinear space. Studying when adversarially robust streaming algorithms are possible is an important problem in lieu of recent interest in adversarial attacks in ML with applications to adaptive data analysis.

Formally, we define the model as a two-player game between a streaming algorithm StreamAlg and a source Adversary of adaptive and adversarial input to StreamAlg. At the beginning of the game, a fixed query $\mathcal{Q}$ is determined and asks for a fixed function for the underlying dataset implicitly defined by the stream. The game then proceeds in rounds, and in the $t$-th round,

(1) Adversary computes an update $u_t \in [n]$ for the stream, which depends on all previous stream updates and all previous outputs from StreamAlg.

(2) StreamAlg uses $u_t$ to update its data structures $D_t$, acquires a fresh batch $R_t$ of random bits, and outputs a response $A_t$ to the query $\mathcal{Q}$.

(3) Adversary observes and records the response $A_t$.

The goal of Adversary is to induce StreamAlg to make an incorrect response $A_t$ to the query $\mathcal{Q}$ at some time $t \in [m]$ throughout the stream.

**Related Works.** Adversarial robustness of streaming algorithms has been an important topic of recent research. On the positive note, [BJWY20] gave a robust framework for estimating the $L_p$ norm of points in a stream in the insertion-only model, where previous stream updates cannot later be deleted. Their work thus shows that deletions are integral to the attack of [HW13]. Subsequently, [HKM+20] introduced a new algorithmic design for robust $L_p$ norm estimation algorithms, by using differential privacy to protect the internal randomness of algorithms against the adversary. Although [WZ20] tightened these bounds, showing that essentially no losses related to the size of the input $n$ or the accuracy parameter $\varepsilon$ were needed, [KMNS21] showed that this may not be true in general. Specifically, they showed a separation between oblivious and adversarial streaming in the adaptive data analysis problem.

[BY20] showed that sampling is not necessarily adversarially robust; they introduce an exponentially sized set system where a constant number of samples, corresponding to the VC-dimension of the set system, may result in a very unrepresentative set of samples. However, they show that with an additional logarithmic overhead in the number of samples, then Bernoulli and or reservoir sampling are adversarially robust. This notion is further formalized by [ABD+21], who showed that the classes that are online learnable requires essentially sample-complexity proportional to the Littlestone's dimension of the underlying set system, rather than VC dimension. However, these sampling procedures are uniform in the sense that each item in the stream is sampled with the same probability. Thus the sampling probability of each item is *oblivious* to the identity of the item. By contrast, we show the robustness for a variety of algorithms based on *non-oblivious* sampling, where each stream item is sampled with probability roughly proportional to the "importance" of the item.

## 1.1 Our Contributions

Our main contribution is a powerful yet simple statement that algorithms based on non-oblivious sampling are adversarially robust if informally speaking, the process of sampling each item in the

stream can be viewed as using fresh randomness independent of previous steps, even if the sampling probabilities depend on previous steps.

Let us describe, very informally, our meta-approach. Suppose we have an adversarial stream of elements given by $u_1, \ldots, u_n$. Our algorithm $\mathcal{A}$ will maintain a data structure $A_t$ at time $t$ which updates as the stream progresses. $\mathcal{A}$ will use a function $g(A_t, u_t)$ to determine the probability of sampling item $u_t$ to update $A_t$ to $A_{t+1}$. The function $g$ measures the 'importance' of the element $u_t$ to the overall problem that we wish to solve. For example, if our application is $k$-means clustering and $u_t$ is a point far away from all previously seen points so far, we want to sample it with a higher probability. We highlight that even though the sampling probability for $u_t$ given by $g(A_t, u_t)$ is adversarial, since the adversary designs $u_t$ and previous streaming elements, the *coin toss* performed by our algorithm $\mathcal{A}$ to keep item $u_t$ is *independent* of any events that have occurred so far, including the adversary's actions. This new randomness introduced by the independent coin toss is a key conceptual step in the analysis for all of the applications listed in Figure 1.

Contrast this to the situation where a "fixed" data structure or sketch is specified upfront. In this case, we would not be adaptive to which inputs $u_t$ the adversary designs to be "important" for our problem which would lead us to potentially disregard such important items rendering the algorithm ineffective.

As applications of our meta-approach, we introduce adversarially robust streaming algorithms for two central machine learning tasks, regression and clustering, as well as their more general counterparts, subspace embedding, low-rank approximation, and coreset construction.

We show that several methods from the streaming algorithms "toolbox", namely merge and reduce, online leverage score sampling, and edge sampling are adversarially robust "for free." As a result, *existing* (and future) streaming algorithms that use these tools are robust as well. We discuss our results in more detail below and provide a summary of our results and applications in Figure 1.

| Meta-approach | Applications |
|---|---|
| Merge and reduce (Theorem 1.1) | Coreset construction, support vector machine, Gaussian mixture models, $k$-means clustering, $k$-median clustering, projective clustering, principal component analysis, $M$-estimators, Bayesian logistic regression, generative adversarial networks (GANs), $k$-line center, $j$-subspace approximation, Bregman clustering |
| Row sampling (Theorem 1.2) | Linear regression, generalized regression, spectral approximation, low-rank approximation, projection-cost preservation, $L_1$-subspace embedding |
| Edge sampling (Theorem 1.3) | Graph sparsification |

Fig. 1: Summary of our robust sampling frameworks and corresponding applications

We first show that the well-known merge and reduce paradigm is adversarially robust. Since the merge and reduce paradigm defines coreset constructions, we thus obtain robust algorithms for $k$-means, $k$-median, Bregman clustering, projective clustering, principal component analysis (PCA), non-negative matrix factorization (NNMF) [LK17].

**Theorem 1.1 (Merge and reduce is adversarially robust)** *Given an offline $\varepsilon$-coreset construction, the merge and reduce framework gives an adversarially robust streaming construction for an $\varepsilon$-coreset with high probability.*

For regression and other numerical linear algebra related tasks, we consider the row arrival streaming model, in which the adversary generates a sequence of row vectors $\mathbf{a}_1, \ldots, \mathbf{a}_n$ in $d$-dimensional vector space. For $t \in [n]$, the $t$-th prefix of the stream induces a matrix $\mathbf{A}_t \in \mathbb{R}^{t \times d}$ with rows $\mathbf{a}_1, \ldots, \mathbf{a}_t$. We denote this matrix as $\mathbf{A}_t = \mathbf{a}_1 \circ \ldots \circ \mathbf{a}_t$ and define $\kappa$ to be an upper bound on the largest condition number[1] of the matrices $\mathbf{A}_1, \ldots, \mathbf{A}_n$.

**Theorem 1.2 (Row sampling is adversarially robust)** *There exists a row sampling based framework for adversarially robust streaming algorithms that at each time $t \in [n]$:*

---

[1] the ratio of the largest and smallest nonzero singular values

(1) *Outputs a matrix* $\mathbf{M}_t$ *such that* $(1-\varepsilon)\mathbf{A}_t^\top\mathbf{A}_t \preceq \mathbf{M}_t^\top\mathbf{M}_t \preceq (1+\varepsilon)\mathbf{A}_t^\top\mathbf{A}_t$, *while sampling* $\mathcal{O}\left(\frac{d^2\kappa}{\varepsilon^2}\log n\log\kappa\right)$ *rows (spectral approximation/subspace embedding/linear regression/generalized regression).*

(2) *Outputs a matrix* $\mathbf{M}_t$ *such that for all rank $k$ orthogonal projection matrices* $\mathbf{P}\in\mathbb{R}^{d\times d}$,

$$(1-\varepsilon)\left\|\mathbf{A}_t-\mathbf{A}_t\mathbf{P}\right\|_F^2 \le \left\|\mathbf{M}_t-\mathbf{M}_t\mathbf{P}\right\|_F^2 \le (1+\varepsilon)\left\|\mathbf{A}_t-\mathbf{A}_t\mathbf{P}\right\|_F^2,$$

*while sampling* $\mathcal{O}\left(\frac{dk\kappa}{\varepsilon^2}\log n\log^2\kappa\right)$ *rows (projection-cost preservation/low-rank approximation).*

(3) *Outputs a matrix* $\mathbf{M}_t$ *such that* $(1-\varepsilon)\left\|\mathbf{A}_t\mathbf{x}\right\|_1 \le \left\|\mathbf{M}_t\mathbf{x}\right\|_1 \le (1+\varepsilon)\left\|\mathbf{A}_t\mathbf{x}\right\|_1$, *while sampling* $\mathcal{O}\left(\frac{d^2\kappa}{\varepsilon^2}\log^2 n\log\kappa\right)$ *rows ($L_1$ subspace embedding).*

Finally, we show that our analysis also applies to algorithms for graph sparsification for in which edges are sampled according to their "importance". Define $\kappa$ as the ratio of the largest and the smallest cut sizes in $G$ (see Section 4 and Supplementary Section C for exact details).

**Theorem 1.3** *Given a weighted graph $G=(V,E)$ with $|V|=n$ whose edges $e_1,\ldots,e_m$ arrive sequentially in a stream, there exists an adversarially robust streaming algorithm that outputs a $1\pm\varepsilon$ cut sparsifier with $\mathcal{O}\left(\frac{\kappa^2 n\log n}{\varepsilon^2}\right)$ edges with probability $1-1/\operatorname{poly}(n)$.*

**Sketching vs Sampling Algorithms.** A central tool for randomized streaming algorithms is the use of linear sketches. These methods maintain a data structure $f$ such that after the $(i+1)$-th input $x_i$, we can update $f$ by computing a linear function of $x_i$. Typically, these methods employ a random matrix. For example, if the input consists of vectors, sketching methods will use a random matrix to project the vector into a much smaller dimension space. In [HW13], it was proved no linear sketch can approximate the $L_2$-norm within a polynomial multiplicative factor against such an adaptive adversary. In general, streaming algorithms that use sketching are highly susceptible to the type of attack described in [HW13] where the adversary can effectively 'learn' the kernel of the linear function used and send inputs along the kernel. For example, if an adversary knows the kernel of the random matrix used to project the input points, then by sending points that lie on the kernel of the matrix as inputs, the adversary can render the whole streaming algorithm useless.

One the other hand, we employ a different family of streaming algorithms that are based on *sampling* the input rather than *sketching* it. Surprisingly, this simple change allows one to automatically get many adversarially robust algorithms either "for free" or *without* new algorithmic overheads. For more information, see Section 1.1. We emphasize that while our techniques are not theoretically sophisticated, we believe its power lies in its simple message that **sampling is often superior to sketching for adversarial robustness**. In addition to downstream algorithmic and ML applications, this provides an interesting separation and trade-offs between the two paradigms; for non adversarial inputs sketching often gives similar or better performance guarantees for many tasks [BYKS01].

## 2 Merge and Reduce

We show that the general merge and reduce paradigm is adversarially robust. Merge and reduce is widely used for the construction of a coreset, which provides dimensionality reduction on the size of an underlying dataset, so that algorithms for downstream applications can run more efficiently:

**Definition 2.1** ($\varepsilon$ coreset) *Let $P\subset X$ be a set of elements from a universe $X$, $z\ge 0$, $\varepsilon\in(0,1)$, and $(P,\operatorname{dist},Q)$ be a query space. Then a subset $C$ equipped with a weight function $w:P\to\mathbb{R}$ is called an $\varepsilon$-coreset with respect to the query space $(P,\operatorname{dist},Q)$ if*

$$(1-\varepsilon)\sum_{\mathbf{p}\in P}\operatorname{dist}(\mathbf{p},Q)^z \le \sum_{\mathbf{p}\in C}w(p)\operatorname{dist}(\mathbf{p},Q)^z \le (1+\varepsilon)\sum_{\mathbf{p}\in P}\operatorname{dist}(\mathbf{p},Q)^z.$$

The study of efficient offline coreset constructions for a variety of geometric and algebraic problems forms a long line of active research. For example, offline coreset constructions are known for linear

regression, low-rank approximation, $L_1$-subspace embedding, $k$-means clustering, $k$-median clustering, $k$-center, support vector machine, Gaussian mixture models, $M$-estimators, Bregman clustering, projective clustering, principal component analysis, $k$-line center, $j$-subspace approximation, and so on. Thus, our result essentially shows that using the merge and reduce paradigm, these offline coreset constructions can be extended to obtain robust and accurate streaming algorithms. The merge and reduce paradigm works as follows. Suppose we have a stream $p_1, \ldots, p_n$ of length $n = 2^k$ for some

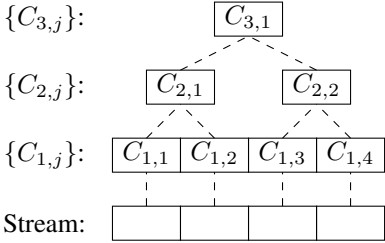

Fig. 2: Merge and reduce framework. Each $C_{1,j}$ is an $O(\varepsilon/\log n)$-coreset of the corresponding partition of the substream and each $C_{i,j}$ is an $\varepsilon$-coreset of $C_{i-1,2j-1}$ and $C_{i-1,2j}$ for $i > 1$.

integer $k$, without loss of generality (otherwise we can use a standard padding argument to increase the length of the stream). Define $C_{0,j} = p_j$ for all $j \in [n]$. Consider $k$ levels, where each level $i \in [k]$ consists of $\frac{n}{2^i}$ coresets $C_{i,1}, \ldots, C_{i,n/2^i}$ and each coreset $C_{i,j}$ is an $\frac{\varepsilon}{2k}$-coreset of $C_{i-1,2j-1}$ and $C_{i-1,2j}$. Note that this approach can be implemented efficiently in the streaming model, since each $C_{i,j}$ can be built immediately once $C_{i-1,2j-1}$ and $C_{i-1,2j}$ are constructed, and after $C_{i,j}$ is constructed, then both $C_{i-1,2j-1}$ and $C_{i-1,2j}$ can be discarded. For an illustration of the merge and reduce framework, see Figure 2, though we defer all formal proofs to the supplementary material. Using the coresets of [BDM$^+$20], Theorem 1.1 gives the following applications:

**Theorem 2.2** *There exists a merge-and-reduce row sampling based framework for adversarially robust streaming algorithms that at each time $t \in [n]$:*

*(1) Outputs a matrix $\mathbf{M}_t$ such that $(1 - \varepsilon)\mathbf{A}_t^\top \mathbf{A}_t \preceq \mathbf{M}_t^\top \mathbf{M}_t \preceq (1 + \varepsilon)\mathbf{A}_t^\top \mathbf{A}_t$, while sampling $\mathcal{O}\left(\frac{d^2}{\varepsilon^2} \log^4 n \log \kappa\right)$ rows (spectral approximation/subspace embedding/linear regression/generalized regression).*

*(2) Outputs a matrix $\mathbf{M}_t$ such that for all rank $k$ orthogonal projection matrices $\mathbf{P} \in \mathbb{R}^{d \times d}$,*
$$(1 - \varepsilon) \|\mathbf{A}_t - \mathbf{A}_t \mathbf{P}\|_F^2 \leq \|\mathbf{M}_t - \mathbf{M}_t \mathbf{P}\|_F^2 \leq (1 + \varepsilon) \|\mathbf{A}_t - \mathbf{A}_t \mathbf{P}\|_F^2,$$
*while sampling $\mathcal{O}\left(\frac{k}{\varepsilon^2} \log^4 n \log^2 \kappa\right)$ rows (projection-cost preservation/low-rank approximation).*

*(3) Outputs a matrix $\mathbf{M}_t$ such that $(1 - \varepsilon) \|\mathbf{A}_t \mathbf{x}\|_1 \leq \|\mathbf{M}_t \mathbf{x}\|_1 \leq (1 + \varepsilon) \|\mathbf{A}_t \mathbf{x}\|_1$, while sampling $\mathcal{O}\left(\frac{d}{\varepsilon^2} \log^5 n \log \kappa\right)$ rows ($L_1$ subspace embedding).*

Using coresets of [HV20], then Theorem 1.1 also gives applications for $(k, z)$-clustering such as $k$-median for $z = 1$ and $k$-means for $z = 2$. Moreover, [LK17] noted that constructions of [FL11] give coresets for Bregman clustering, which handles $\mu$-similar Bregman divergences such as the Itakura-Saito distance, KL-divergence, Mahalanobis distance, etc.

**Theorem 2.3** *There exists a merge-and-reduce importance sampling based framework for adversarially robust streaming algorithms that at each time $t$:*

*(1) Outputs a set of centers that gives a $(1 + \varepsilon)$-approximation to the optimal $(k, z)$-clustering, $k$-means clustering ($z = 2$), and $k$-median clustering ($z = 1$), while storing $\mathcal{O}\left(\frac{1}{\varepsilon^{2z+2}} k \log^{2z+2} n \log k \log \frac{k \log n}{\varepsilon}\right)$ points.*

*(2) Outputs a set of centers that gives a $(1 + \varepsilon)$-approximation to the optimal $k$-Bregman clustering, while storing $\mathcal{O}\left(\frac{1}{\varepsilon^2} dk^3 \log^3 n\right)$ points.*

Using the sensitivity bounds of [VX12a, VX12b] and the coreset constructions of [BFL16], then Theorem 1.1 also gives applications for the following shape fitting problems:

**Theorem 2.4** *There exists a merge-and-reduce importance sampling based framework for adversarially robust streaming algorithms that at each time $t$:*

*(1) Outputs a set of lines that gives a $(1 + \varepsilon)$-approximation to the optimal $k$-lines clustering, while storing $\mathcal{O}\left(\frac{d}{\varepsilon^2}\, f(d,k) k^{f(d,k)} \log^4 n\right)$ points of $\mathbb{R}^d$, for a fixed function $f(d,k)$.*

*(2) Outputs a subspace that gives a $(1 + \varepsilon)$-approximation to the optimal dimension $j$ subspace approximation, while storing $\mathcal{O}\left(\frac{d}{\varepsilon^2}\, g(d,j) k^{g(d,j)} \log^4 n\right)$ points of $\mathbb{R}^d$, for a fixed function $g(d,j)$.*

*(3) Outputs a set of subspaces that gives a $(1+\varepsilon)$-approximation to the optimal $(j,k)$-projective clustering, while storing $\mathcal{O}\left(\frac{d}{\varepsilon^2}\, h(d,j,k) \log^3 n (\log n)^{h(d,j,k)}\right)$ points of $\mathbb{R}^d$s, for a fixed function $h(d,j,k)$, for a set of input points with integer coordinates.*

Adversarially robust approximation algorithms for Bayesian logistic regression, Gaussian mixture models, generative adversarial networks (GANs), and support vector machine can be obtained from Theorem 1.1 and coreset constructions of [HCB16, FKW19, SZG$^+$20, TBFR20]; a significant number of additional applications of Theorem 1.1 using coreset constructions can be seen from recent surveys on coresets, e.g., see [LK17, Fel20]. The merge-and-reduce framework also has applications to a large number of other problems such as finding heavy-hitters [MG82] or frequent directions [GLPW16] and in various settings, such as the sliding window model [DGIM02], time decay models [BLUZ19] or for at-the-time or back-in-time queries [SZP$^+$21].

## 3 Adversarial Robustness of Subspace Embedding and Applications

We use $[n]$ to represent the set $\{1, \ldots, n\}$ for an integer $n > 0$. We typically use bold font to denote vectors and matrices. For a matrix $\mathbf{A}$, we use $\mathbf{A}^{-1}$ to denote the Moore-Penrose inverse of $\mathbf{A}$. We first formally define the goals of our algorithms:

**Problem 3.1 (Spectral Approximation)** *Given a matrix $\mathbf{A} \in \mathbb{R}^{n \times d}$ and an approximation parameter $\varepsilon > 0$, the goal is to output a matrix $\mathbf{M} \in \mathbb{R}^{m \times d}$ with $m \ll n$ such that $(1 - \varepsilon) \|\mathbf{Ax}\|_2 \le \|\mathbf{Mx}\|_2 \le (1 + \varepsilon) \|\mathbf{Ax}\|_2$ for all $\mathbf{x} \in \mathbb{R}^d$ or equivalently, $(1-\varepsilon)\mathbf{A}^\top \mathbf{A} \preceq \mathbf{M}^\top \mathbf{M} \preceq (1+\varepsilon)\mathbf{A}^\top \mathbf{A}$.*

We note that linear regression is a well-known specific application of spectral approximation.

**Problem 3.2 (Projection-Cost Preservation)** *Given a matrix $\mathbf{A} \in \mathbb{R}^{n \times d}$, a rank parameter $k > 0$, and an approximation parameter $\varepsilon > 0$, the goal is to find a matrix $\mathbf{M} \in \mathbb{R}^{m \times d}$ with $m \ll n$ such that for all rank $k$ orthogonal projection matrices $\mathbf{P} \in \mathbb{R}^{d \times d}$,*

$$(1 - \varepsilon) \|\mathbf{A} - \mathbf{AP}\|_F^2 \le \|\mathbf{M} - \mathbf{MP}\|_F^2 \le (1 + \varepsilon) \|\mathbf{A} - \mathbf{AP}\|_F^2.$$

Note if $\mathbf{M}$ is a projection-cost preservation of $\mathbf{A}$, then its best low-rank approximation can be used to find a projection matrix that gives an approximation of the best low-rank approximation to $\mathbf{A}$.

**Problem 3.3 (Low-Rank Approximation)** *Given a matrix $\mathbf{A} \in \mathbb{R}^{n \times d}$, a rank parameter $k > 0$, and an approximation parameter $\varepsilon > 0$, find a rank $k$ matrix $\mathbf{M} \in \mathbb{R}^{n \times d}$ such that $(1 - \varepsilon) \left\|\mathbf{A} - \mathbf{A}_{(k)}\right\|_F^2 \le \|\mathbf{A} - \mathbf{M}\|_F^2 \le (1 + \varepsilon) \left\|\mathbf{A} - \mathbf{A}_{(k)}\right\|_F^2$, where $\mathbf{A}_{(k)}$ for a matrix $\mathbf{A}$ denotes the best rank $k$ approximation to $\mathbf{A}$.*

**Problem 3.4 ($L_1$-Subspace Embedding)** *Given a matrix $\mathbf{A} \in \mathbb{R}^{n \times d}$ and an approximation parameter $\varepsilon > 0$, the goal is to output a matrix $\mathbf{M} \in \mathbb{R}^{m \times d}$ with $m \ll n$ such that $(1 - \varepsilon) \|\mathbf{Ax}\|_1 \le \|\mathbf{Mx}\|_1 \le (1 + \varepsilon) \|\mathbf{Ax}\|_1$ for all $\mathbf{x} \in \mathbb{R}^d$.*

We consider the general class of row sampling algorithms, e.g., [CMP16, BDM$^+$20]. Here we maintain a $L_p$ subspace embedding of the underlying matrix by approximating the online $L_p$ sensitivities of each row as a measure of importance to perform sampling. For more details, see Algorithm 1.

**Definition 3.5 (Online $L_p$ Sensitivities)** *For a matrix $\mathbf{A} = \mathbf{a}_1 \circ \ldots \circ \mathbf{a}_n \in \mathbb{R}^{n \times d}$, the online sensitivity of row $\mathbf{a}_i$ for each $i \in [n]$ is the quantity $\max_{\mathbf{x} \in \mathbb{R}^d} \frac{|\langle \mathbf{a}_i, \mathbf{x}\rangle|^p}{\|A_i \mathbf{x}\|_p^p}$, where $\mathbf{A}_{i-1} = \mathbf{a}_1 \circ \ldots \circ \mathbf{a}_{i-1}$.*

---

**Algorithm 1** Row sampling based algorithms, e.g., [CMP16, BDM$^+$20]

---

**Input:** A stream of rows $\mathbf{a}_1, \ldots, \mathbf{a}_n \in \mathbb{R}^d$, parameter $p > 0$, and an accuracy parameter $\varepsilon > 0$
**Output:** A $(1 + \varepsilon)$ $L_p$ subspace embedding.
 1: $\mathbf{M} \leftarrow \emptyset$
 2: $\alpha \leftarrow \frac{Cd}{\varepsilon^2} \log n$ with sufficiently large parameter $C > 0$
 3: **for** each row $\mathbf{a}_i$, $i \in [n]$ **do**
 4:     **if** $\mathbf{a}_i \in \mathrm{span}(\mathbf{M})$ **then**
 5:         $\tau_i \leftarrow 2 \cdot \max_{\mathbf{x} \in \mathbb{R}^d, \mathbf{x} \in \mathrm{span}(\mathbf{M})} \frac{|\langle \mathbf{a}_i, \mathbf{x} \rangle|^p}{\|\mathbf{M}\mathbf{x}\|_p^p + |\langle \mathbf{a}_i, \mathbf{x} \rangle|^p}$     ▷See Remark 3.6
 6:     **else**
 7:         $\tau_i \leftarrow 1$
 8:     $p_i \leftarrow \min(1, \alpha \tau_i)$
 9:     With probability $p_i$, $\mathbf{M} \leftarrow \mathbf{M} \circ \frac{\mathbf{a}_i}{p_i^{1/p}}$     ▷Online sensitivity sampling
10: **return** $\mathbf{M}$

---

We remark on standard computation or approximation of the online $L_p$ sensitivities, e.g., see [CEM$^+$15, CMP16, CMM17, BDM$^+$20].

**Remark 3.6** *We note that for $p = 1$, a constant fraction approximation to any online $L_p$ sensitivity $\tau_i$ such that $\tau_i > \frac{1}{\mathrm{poly}(n)}$ can be computed in polynomial time using (offline) linear programming while for $p = 2$, $\tau_i$ is equivalent to the online leverage score of $\mathbf{a}_i$, which has the closed form expression $\mathbf{a}_i^\top (\mathbf{A}_i^\top \mathbf{A}_i)^{-1} \mathbf{a}_i$, which can be approximated by $\mathbf{a}_i^\top (\mathbf{M}^\top \mathbf{M})^{-1} \mathbf{a}_i$, conditioned on $\mathbf{M}$ being a good approximation to $\mathbf{A}_{i-1}$ when $\mathbf{a}_i$ is in the span of $\mathbf{M}$. Otherwise, $\tau_i$ takes value 1 when $\mathbf{a}_i$ is not in the span of $\mathbf{M}$.*

**Lemma 3.7 (Adversarially Robust $L_p$ Subspace Embedding and Linear Regression)** *Given $\varepsilon > 0$, $p \in \{1, 2\}$, and a matrix $\mathbf{A} \in \mathbb{R}^{n \times d}$ whose rows $\mathbf{a}_1, \ldots, \mathbf{a}_n$ arrive sequentially in a stream with condition number at most $\kappa$, there exists an adversarially robust streaming algorithm that outputs a $(1 + \varepsilon)$ spectral approximation with high probability. The algorithm samples $\mathcal{O}\left( \frac{d^2 \kappa^2}{\varepsilon^2} \log n \log \kappa \right)$ rows for $p = 2$ and $\mathcal{O}\left( \frac{d^2 \lambda^2}{\varepsilon^2} \log^2 n \log \kappa \right)$ rows for $p = 1$, with high probability, where $\lambda$ is a ratio between upper and lower bounds on $\|\mathbf{A}\|_1$.*

We also show robustness of row sampling for low-rank approximation by using online ridge-leverage scores. Together, Lemma 3.7 and Lemma 3.8 give Theorem 1.2.

**Lemma 3.8 (Adversarially Robust Low-Rank Approximation)** *Given accuracy parameter $\varepsilon > 0$, rank parameter $k > 0$, and a matrix $\mathbf{A} \in \mathbb{R}^{n \times d}$ whose rows $\mathbf{a}_1, \ldots, \mathbf{a}_n$ arrive sequentially in a stream with condition number at most $\kappa$, there exists an adversarially robust streaming algorithm that outputs a $(1 + \varepsilon)$ low-rank approximation with high probability. The algorithm samples $\mathcal{O}\left( \frac{kd\kappa^2}{\varepsilon^2} \log n \log \kappa \right)$ rows with high probability.*

## 4 Graph Sparsification

In this section, we highlight how the sampling paradigm gives rise to an adversarially robust streaming algorithm for graph sparsification. First, we motivate the problem of graph sparsification. Massive graphs arise in many theoretical and applied settings, such as in the analysis of large social or biological networks. A key bottleneck in such analysis is the large computational resources, in both memory and time, needed. Therefore, it is desirable to get a representation of graphs that take up far less space while still preserving the underlying "structure" of the graph. Usually the number of vertices is much fewer than the number of edges; for example in typical real world graphs, the number of vertices can be several orders of magnitude smaller than the number of edges (for example, see the graph datasets in [RA15]). Hence, a natural benchmark is to reduce the number of edges to be comparable to the number of vertices.

The most common notion of graph sparsification is that of preserving the value of *all* cuts in the graph by keeping a small weighted set of edges of the original graph. More specifically, suppose our

graph is $G = (V, E)$ and for simplicity assume all the edges have weight 1. A cut of the graph is a partition of $V = (C, V \setminus C)$ and the value of a cut, $\text{Val}_G(C)$, is defined as the number of edges that cross between the vertices in $C$ and $V \setminus C$. A graph $H$ on the same set of vertices as $V$ is a sparsifier if it preserves the value of every cut in $G$ and has a few number of weighted edges. For a precise formulation, see Problem 4.1.

In addition to being algorithmically tractable, this formulation is natural since it preserves the underlying cluster structure of the graph. For example, if there are two well connected components separated by a sparse cut, i.e. two distinct communities, then the sparsifier according to the definition above will ensure that the two communities are still well separated. Conversely, by considering any cut within a well connected component, it will also ensure that any community remains well connected (for more details, see [SPR11] and references therein). Lastly, graph sparsification has been considered in other frameworks such as differential privacy [EKKL20], distributed optimization [WWLZ18], and even learning graph sparsification using deep learning methods [ZZC+20]. The formal problem definition of graph sparsification is as follows.

**Problem 4.1 (Graph Sparsification)** *Given a graph weighted $G = (V, E)$ with $|V| = n, |E| = m$, and an approximation parameter $\varepsilon > 0$, compute a weighted subgraph $H$ of $G$ on the same set of vertices such that*

*(1) every cut in $H$ has value between $1 - \varepsilon$ and $1 + \varepsilon$ times its value in $G$: $(1 - \varepsilon)\text{Val}_G(C) \leq \text{Val}_H(C) \leq (1 + \varepsilon)\text{Val}_G(C)$ for all cuts $C$ where $\text{Val}_G(C), \text{Val}_H(C)$ denotes the cost of the cut in the graphs $G$ and $H$ respectively and for the latter quantity, the edges are weighted,*

*(2) the number of edges in $H$ is $\mathcal{O}\left(\frac{n \log n}{\varepsilon^2}\right)$.*

Ignoring dependence on $\varepsilon$, there are previous results that already get sparsifiers $H$ with $O(n \log n)$ edges [BK96, SS08]. Their setting is when the *entire* graph is present up-front in memory. In contrast, we are interested in the streaming setting where future edges can depend on past edges as well as revealed randomness of an algorithm while processing the edges.

Our main goal is to show that the streaming algorithm from [AG09] (presented in Algorithm 2 in the supplementary section), which uses a sampling procedure to sample edges in a stream, is adversarially robust, albeit with a slightly worse guarantee for the number of edges. Following the proof techniques of the non streaming algorithm given in [BK96], it is shown in [AG09] that Algorithm 2 outputs a subgraph $H$ such that $H$ satisfies the conditions of Problem 4.1 with probability $1 - 1/\text{poly}(n)$ where the probability can be boosted by taking a larger constant $C$. We must show that this still holds true if the edges of the stream are adversarially chosen, i.e., when **new edges in the stream depend on the previous edges and the randomness used by the algorithm so far.** We thus again use a martingale argument; the full details are given in Supplementary Section C. As in Section 3, we let $\kappa_1$ and $\kappa_2$ to be deterministic lower/upper bounds on the size of any cut in $G$ and define $\kappa = \kappa_2/\kappa_1$.

**Theorem 1.3** *Given a weighted graph $G = (V, E)$ with $|V| = n$ whose edges $e_1, \ldots, e_m$ arrive sequentially in a stream, there exists an adversarially robust streaming algorithm that outputs a $1 \pm \varepsilon$ cut sparsifier with $\mathcal{O}\left(\frac{\kappa^2 n \log n}{\varepsilon^2}\right)$ edges with probability $1 - 1/\text{poly}(n)$.*

## 5 Experiments

To illustrate the robustness of importance-sampling-based streaming algorithms we devise adversarial settings for clustering and linear regression. With respect to our adversarial setting, we show that the performance of a merge-and-reduce based streaming $k$-means algorithm is robust while a popular streaming $k$-means implementation (not based on importance sampling) is not. Similarly, we show the robustness superiority of a streaming linear regression algorithm based on row sampling over a popular streaming linear regression implementation and over sketching.

**Streaming $k$-means** In this adversarial clustering setting we consider a series of point batches where all points except those in the last batch are randomly sampled from a two dimensional standard normal distribution and points in the last batch similarly sampled but around a distant center (see the data points realization in both panels of Figure 3). We then feed the point sequence

to `StreamingKMeans`, the streaming $k$-means implementation of Spark [ZXW$^+$16b] the popular big-data processing framework. As illustrated in the left panel of Figure 3, the resulting two centers are both within the origin. Now, this result occurs regardless of the last batch's distance from the origin, implying that the per-sample loss performance of the algorithm can be made arbitrarily large. Alternatively, we used a merge-and-reduce based streaming $k$-means algorithm and show that one of the resulting cluster centers is at the distant cluster (as illustrated in the right panel of Figure 3) thereby keeping the resulting per sample loss at the desired minimum. Specifically we use `Streamkm`, an implementation of StreamKM++ [AMR$^+$12] from the ClusOpt Core library [Mac20].

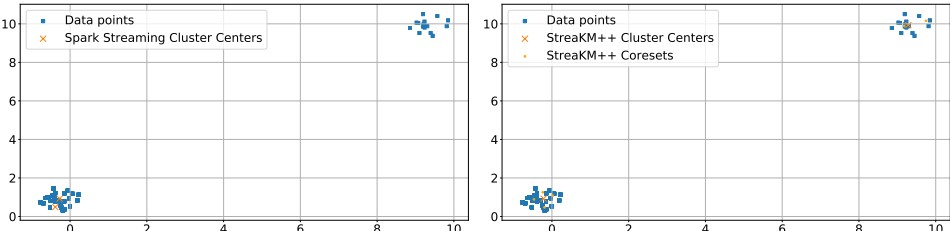

Fig. 3: Resulting cluster centers (x) on our adversarial datapoints setting for the popular Spark implementation (left) and StreamKM++ (right).

**Streaming linear regression.** Similar to the clustering setting, in the adversarial setting for streaming linear regression all batches except the last one are sampled around a constellation of four points in the plane such that the optimal regression line is of $-1$ slope through the origin (see the leftmost panel of Figure 4). The last batch of points however, is again far from the origin $(L, L)$ such that the resulting optimal regression line is of slope 1 through the origin[2]. We compare the performance of `LinearRegression` from the popular streaming machine learning library River [MHM$^+$20] to our own row sampling based implementation of streaming linear regression along the lines of Algorithm 1 and observe the following: Without the last batch of points, both implementations result in the optimal regression line, however, the River implementation reaches that line only after several iterations, while our implementation is accurate throughout (This is illustrated in the second-left panel of Figure 4). When the last batch is used, nevertheless, Algorithm 1 picks up the drastic change and adapts immediately to a line of the optimal slope (the blue line of the second right panel of Figure 4) while the River implementation update merely moves the line in the desired direction (the orange line in that same panel) but is far from catching up. Finally, the rightmost panel of Figure 4) details the loss trajectory for both implementations. While the River loss skyrockets upon the last batch, the loss of Algorithm 1 remains relatively unaffected, illustrating its adversarial robustness.

Note that in both the clustering and linear regression settings above the adversary was not required to consider the algorithms internal randomization to achieve the desired effect (this is due to the local nature of the algorithms computations). This is no longer the case in the following last setting.

**Sampling vs. sketching.** Finally, we compare the performance of the leverage sampling Algorithm 1 to sketching. In this setting, for a random unit sketching matrix $S$ (that is, each of its elements is sampled from $\{-1, 1\}$ with equal probability), we create an adversarial data stream $A$ such that its columns are in the null space of $S$. As a result, the linear regression as applied to the sketched data $S \cdot A$ as a whole is unstable and might significantly differ from the resulting linear regression applied to streamed prefixes of the sketched data. As illustrated in Figure 5, this is not the case when applying the linear regression to the original streamed data $A$ using Algorithm 1. Upon the last batch, the performance of the sketching-based regression deteriorates by orders of magnitude, while the performance of Algorithm 1 is not affected. Moreover, the data reduction factor achieved by leveraged sampling[3] is almost double compared to the data reduction factor achieved by sketching.

## Acknowledgments

Sandeep Silwal was supported in part by a NSF Graduate Research Fellowship Program. Samson Zhou was supported by a Simons Investigator Award of David P. Woodruff.

---

[2]For MSE loss, this occurs for $L$ at least the square root of the number of batches.

[3]The original stream $A$ contained 2000 samples, each of dimension 10.

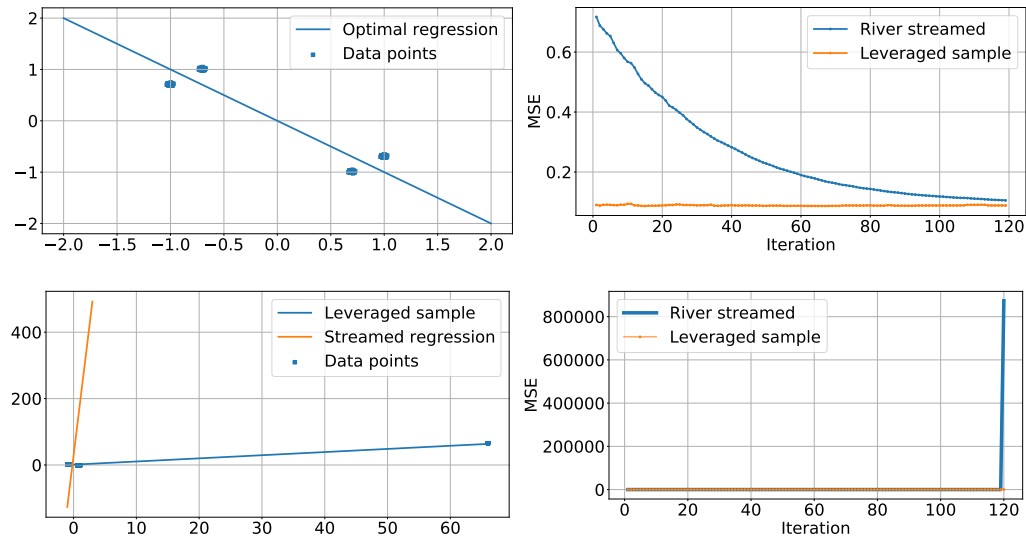

Fig. 4: Streaming linear regression experiment (from left to right): points constellation without last batch, loss trajectory up to (not including) last batch, resulting regression lines upon training with last batch, and loss trajectory including the last batch.

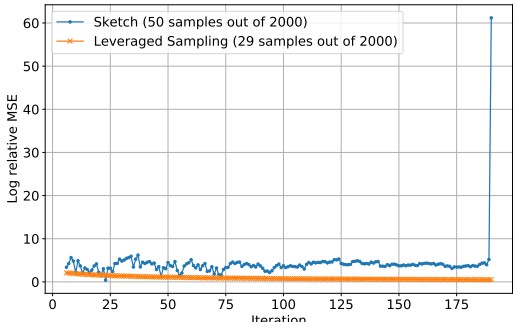

Fig. 5: Comparing the performance trajectory of leveraged sampling Algorithm 1 to sketching, for an adversarial data stream tailored to a sketching matrix. Sketching performance deteriorates catastrophically upon the last batch, while leveraged sampling remains robust.

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
