# OpenReview forum: "Adversarial Robustness of Streaming Algorithms through Importance Sampling"
_NeurIPS.cc/2021/Conference — NeurIPS 2021 Poster_

### Official Review · Reviewer_AQVy · 2021-07-10

**Rating:** 7
**Confidence:** 3

**Summary:**

This paper show how streaming algorithms based on importance sampling can achieve adversarial robustness in a specific but quite general adversarial model, where the adversary observes all the intermediate outputs of the algorithm and may even learn past random bits, but has no access to the "new" random bits used at each time. Any method that uses importance sampling (in the way described in the paper) is adversarial-robust, and many existing streaming approaches enjoy this property e.g., , those using coresets and the merge-and-reduce approach, and methods for regression using row sampling and methods for graph sparsification using edge sampling.


**Limitations And Societal Impact:**

There is no discussion of *algorithmic* limitations of the work, although the paper would benefit from such a discussion.

This Reviewer does not see any negative societal impact that is *only specific* to this work.

**Main Review:**

## Comments after response

I thank the Authors for their response. I read it carefully, as I did the comments from other Reviewers and the responses to them. I believe should be accepted, and I increased my original score to reflect this new opinion. The new figures look good and should be included in the final version of the paper in place of the original ones.

## Original Review

The results in this paper seem quite interesting. After [HV20], it is nice to see more work remarking on the beneficial properties of importance sampling, in this case versus "non-adaptive" sketching.

One downside of the paper is that the main text contains no proofs at all, not even sketches, roadmaps, intuitions, informal ideas, or anything that may guide the reader, who may not want to delve into the whole supplementary materials, into understanding *why* the proposed results are true. Indeed the paper, due to space restrictions, reads almost as just a list of results, which is a pity. I wonder whether the Authors could avoid presenting some results (deferring those to the supplementary materials), and present more discussion and possibly some proof sketches.

The experimental evaluation shows the large difference between adversarially-robust and "non-robust" methods. The settings and experiments feel a bit artificial, but they do drive home the point.

The figures and their discussion make reference to colors and other properties that are lost when the paper is printed in grayscale, and also to colorblind people. Additionally, some figures (e.g., Fig. 4) are too small to be readable when the paper is printed. Using different line styles, bigger markers / thicker lines, and other visual clues would be beneficial.

This Reviewer did not check all the details of all the proofs, but the general "flow" of the proofs seems correct.

**Time Spent Reviewing:**

3

---

> ### Author Response · Authors · 2021-08-10
> **Response to Reviewer AQVy**
>
> Thank you for the comments. We are happy that you found the results in the paper interesting. We have made an editorial pass to incorporate some of the presentational suggestions you made. In particular:
>
> > The figures and their discussion make reference to colors and other properties that are lost when the paper is printed in grayscale, and also to colorblind people. Additionally, some figures (e.g., Fig. 4) are too small to be readable when the paper is printed. Using different line styles, bigger markers / thicker lines, and other visual clues would be beneficial.
>
> We will certainly update our figures so that they are more accessible to a general population, e.g., colorblind people. In particular, we will use larger figures that contain different marker styles for each dataset in a plot.
>
> > The settings and experiments feel a bit artificial, but they do drive home the point.
>
> We remark that in our experimental setup evaluating the popular Spark implementation for $k$-means and River library for linear regression, only the last batch of the data is generated from a different distribution than the previous batches. Thus, the adversary is severely restricted in our setting. Nevertheless, even against such a weak adversary, our experiments show that the popular streaming libraries are heavily impacted by adversarial inputs, whereas algorithms based on importance sampling are adversarially robust. Moreover, we remark that a natural source of this data distribution results from the simple case in which a generally uniform signal receives a burst of noise at the end of the stream.
>
> > One downside of the paper is that the main text contains no proofs at all, not even sketches, roadmaps, intuitions, informal ideas, or anything that may guide the reader, who may not want to delve into the whole supplementary materials, into understanding why the proposed results are true. Indeed the paper, due to space restrictions, reads almost as just a list of results, which is a pity.
>
> We agree that additional intuition would be beneficial for why sampling algorithms can be adversarially robust in contrast to sketching algorithms. In Appendix A, we state that the merge-and-reduce framework is intuitively adversarially robust because “the algorithm has a fresh source of randomness at each time” which is independent of previous randomness and thus also independent of the adversarial input. We provide similar intuition in Appendix B and Appendix C for numerical linear algebra and graph algorithms, respectively. In the case that our paper is accepted, we will move this intuition on the correctness of the algorithms to the main body, using the additional content page that is granted for the camera-ready version. We will also expand the discussion on the challenges for why existing analysis does not work for proving adversarial robustness for these algorithms. Finally, we will add a roadmap detailing the organization of our paper.
>
> We thank you again for your comments. If you feel your main concerns have been addressed, we hope you will consider raising your score. Otherwise, we would also be happy to clarify any misconceptions we may have about your concerns.

---

> > ### Author Response · Authors · 2021-08-21
> > **Follow-up to response to Reviewer AQVy**
> >
> > Dear Reviewer AQVy,
> >
> > Did we address all your concerns satisfactorily? If so, could you please consider raising your score appropriately? If not, could you please let us know which concerns were not sufficiently addressed so that we have a chance to respond? Thanks!

---

> ### Author Response · Authors · 2021-08-25
> **Second Follow-up to response to Reviewer AQVy**
>
> Hi Reviewer AQVy,
>
> Just wanted to check -- would you like to see the updated versions of our figures that are larger and contain different marker styles, to increase accessibility to a larger population, e.g., colorblind people?
>
> More generally, did we address all your concerns satisfactorily? If so, could you please consider raising your score appropriately? If not, would you be able to let us know which remaining reservations you may have, so that we have a chance to respond? Thanks!

---

> > ### Comment · Reviewer_AQVy · 2021-08-31
> > **Thank you!**
> >
> > Thank you for creating new figures and addressing my concerns.

---

> ### Author Response · Authors · 2021-08-30
> **Increased Figures for Visibility**
>
> For concreteness, we wanted to present our figures that have been updated with broader accessibility in mind.
>
> We first recreated the right-hand image of Figure 3 with the dot of the streamKM++ Coresets color changed from blue to orange. Although it has a different marker, it was still difficult to see, so we also increased the size of the figure, which is available below.
>
> https://ibb.co/Q9NY9tH (Updated Figure 3)
>
> We also increased the size of the four figures in Figure 4 to improve visibility.
>
> https://ibb.co/Db8vqVr (Larger Figure 4a)
>
> https://ibb.co/GHc8MHF (Larger Figure 4b)
>
> https://ibb.co/XDMPDJb (Larger Figure 4c)
>
> https://ibb.co/bFZw1Qf (Larger Figure 4d)
>
> Finally, we added different markers in Figure 5, replacing the dots with X marks in the "leveraged sampling line". This was still a little difficult to see, so we also increased the size of the figure.
>
> https://ibb.co/K6z4DTc (Updated Figure 5)
>
> We hope these figures resolve your concerns about the visibility of the figures. If our previous response has also addressed your other reservations, we hope you will consider raising your score. Otherwise, please let us know what remaining reservations you may have, thanks!

---

### Official Review · Reviewer_m5zL · 2021-07-16

**Rating:** 6
**Confidence:** 4

**Summary:**

This paper studies adversarial robust streaming algorithms. In the streaming model, there are many randomized algorithms. An adversary gives a sequence of update to the algorithm adaptively to learn the random bits used by the outputs of the algorithm. A adversarial robust streaming algorithm is robust to any adversarial updates. This paper gives an observation that if the streaming algorithm is sampling based and the random bit for each item is fresh, then the algorithm is robust even though the sampling probability depends on the previous random bits.

**Limitations And Societal Impact:**

I did not find any potential negative societal impact of the paper.

**Main Review:**

Pros:
-Adversarial robustness streaming algorithms are important in the literature of streaming algorithm. Based on the observations of this paper, many previous streaming algorithms automatically are adversarially robust. These problems include many fundamental ML, numerical linear algebra and graph problems including clustering, Gaussian mixture models, SVM, k-clustering, regression, GANs, low rank approximation, spectral approximation, graph scarification, and etc.
-The paper is well-written in general.
-The experiments support their results.

Cons:
-My major concern is the technical strength of the paper. Although the paper obtains many important results, all of them directly follows from previous streaming algorithms. Although authors show that Merge-and-Reduce framework and row sampling framework are adversarially robust in general. The techniques for analysis are not hard. Basically the structures of these frameworks make the analysis works in some sort of a straightforward way.

Overall, since results are important but the technical strength is limited, I think the paper is marginally above the bar.

**Time Spent Reviewing:**

3 hours

---

> ### Author Response · Authors · 2021-08-10
> **Response to Reviewer m5zL**
>
> Thank you for the review. We are glad to hear that you found the problem wel-motiviated and the presentation well-written.
>
> > My major concern is the technical strength of the paper. Although the paper obtains many important results, all of them directly follows from previous streaming algorithms.
>
> We agree that many techniques generalize in a straightforward way, such as the proofs for the merge and reduce framework. However, we would like to emphasize that due to the adversarial nature of the input, a large body of existing techniques no longer work for other settings we consider. For example, many existing algorithms consider a martingale procedure to analyze correctness of the algorithms. However, because we have a random sequence of inputs (which can depend on previous elements and the randomness used by the algorithm), we cannot apply such techniques. For instance, several of the randomized numerical linear algebra algorithms in the oblivious setting are proven using matrix Freedman inequality, but we cannot use this argument in our setting due to the absence of a fixed input.
>
> As a high level overview, previous analysis for streaming algorithms works as follows: they derive two quantities, say $A$ and $B$, where $A$ is the data structure returned by the algorithm, and $B$ is the true underlying quantity defined by the entire stream. For example, $A$ can be a sketch of a matrix and $B$ is the entire matrix which we do not observe. Then, one can compare how well $A$ approximates $B$ using probabilistic tools since $A$ is random but $B$ is a fixed object. However in random and adversarial settings, both $A$ and $B$ can be random quantities and furthermore, $B$ can depend on $A$. This invalidates many of the tools, such as martingale arguments, used in literature. Thus we believe that our results require additional technical work beyond the previous analysis of the existing streaming algorithms. In particular, we prove that $B$ must satisfy certain properties even though it can be a random variable that depends on $A$. Please see the proofs of Lemmas B.2, B.3 as well as the proof of Theorem 1.3 in the supplementary section for more details. Finally, we remark that it is somewhat surprising to us that despite the more complicated analysis, we need not change any aspects of existing algorithms that are based on importance sampling. Thus libraries that implement the importance sampling framework can be used to handle adversarial input with additional algorithmic modifications.
>
> We hope our response has adequately addressed your concerns; if not, we would be happy to engage in a more thorough discussion. Otherwise, we hope that you will consider raising your score if there are no further concerns.

---

> > ### Author Response · Authors · 2021-08-21
> > **Follow-up to response to Reviewer m5zL**
> >
> > Dear Reviewer m5zL,
> >
> > Did we address all your concerns satisfactorily? If so, could you please consider raising your score appropriately? If not, could you please let us know which concerns were not sufficiently addressed so that we have a chance to respond? Thanks!

---

> > > ### Author Response · Authors · 2021-08-25
> > > **Second Follow-up to response to Reviewer m5zL**
> > >
> > > Hi Reviewer m5zL,
> > >
> > > Just wanted to check -- did we address all your concerns satisfactorily? If so, could you please consider raising your score appropriately? If not, would you be able to let us know which remaining reservations you may have, so that we have a chance to respond? Thanks!

---

> > > > ### Comment · Reviewer_m5zL · 2021-08-30
> > > > **Thanks for the response**
> > > >
> > > > Dear authors,
> > > >
> > > > Thank you for the response. My concerns are addressed. I admitted that there are some novelties in the analysis. It would be good to emphasize some of the high level analyzing ideas in the main body of the paper. A brief discussion of why previous analysis fails is also appreciated. Overall, I will keep the score the same, and I hope the presentation can be improved in the camera ready (or the next) version.

---

### Official Review · Reviewer_K1tg · 2021-07-19

**Rating:** 7
**Confidence:** 3

**Summary:**

Many streaming algorithms developed during the last decades are randomized and have an expected solution guarantee (approximation guarantee).

However, the promised guarantee is not with respect to an adaptive adversary that can choose the remaining part of the instance as a function of the decisions taken so far of the algorithm (and thus as a function of the random bits). This has prompted recent work on "adversarial robustness of streaming" to obtain guarantees regarding these stronger adaptive adversaries. First it has been shown that sketching based algorithms (i.e., sample a dimension-reduction matrix before the stream) is not robust against adaptive adversaries. This is because, the adversary can figure out the used matrix and then feed a worst-case instance.

The present paper has a more positive message. They show that a large family of streaming algorithms are in fact also robust against adaptive adversaries. On a high level they show that sampling based algorithms that roughly work as follows: before the arrival of an element e, we calculate a threshold for taking that element e based on the current state of the algorithm (and thus already used randomness). However, we use *new* randomness for deciding wether to take the element or not.  Such sampling based algorithms have been heavily used for e.g. clustering, graph sparsification and regression.

The main result of the paper is this observation together with tons of applications of prior algorithms that the authors list.

**Limitations And Societal Impact:**

The experiments is not really "complete" but the main focus of the paper is theory.

**Main Review:**

The paper is original in that it studies a stronger adversary, makes the insight that certain kind of sampling-based algorithms are robust against this stronger type of adversaries.

Then the paper is less original in the sense that it feels like a long list of applications where the authors basically has verified that the known algorithms "pass the test." (Personally, I'd prefer if they would have focused on one or few applications in the main body of the paper and showed some nice insights there. But if I understand the authors correctly the main result is this important insight (point of view) and then all the results basically follows from previous work.)

Otherwise, I would say that the paper reads well and is clear (although they don't define all the problems that they list). I also think it is a good paper in terms of significance in the sense that it is good to know that all these algorithms are in fact robust. So I think the paper is definitely valuable to the research community.

A more detailed comment to the authors: How important is it for your algorithms to know the length of the stream? It seems pretty crucial when you take a union bound over error probabilities but maybe I am missing something? If not, it would be good with a discussion if it is really necessary to know the length of the stream...

**Time Spent Reviewing:**

4

---

> ### Author Response · Authors · 2021-08-10
> **Response to Reviewer K1tg**
>
> Thank you for the feedback. Our algorithm does not need to know the length of the stream in advance since we require the failure probability to be inversely polynomial in the universe size. Thus it suffices for the stream length to simply be bounded by some large polynomial in the universe size. This seems to be a standard assumption for streaming algorithms, since if the stream length can be arbitrarily large, then functions on the underlying dataset represented by the stream may have a large value that requires a large amount of space just to represent. We will revise the next version of the paper with a discussion on why knowing the streaming length in advance is not necessary.

---

### Author Response · Authors · 2021-08-10
**Thanks to all reviewers**

We thank the reviewers for their thoughtful comments and valuable feedback. We especially appreciate the positive remarks, such as

* the paper reads well and is clear (Reviewer K1tg)
* it is a good paper in terms of significance in the sense that it is good to know that all these algorithms are in fact robust (Reviewer K1tg)
* the paper is definitely valuable to the research community (Reviewer K1tg)
* adversarial robustness streaming algorithms are important in the literature of streaming algorithms (Reviewer m5zL)
* the paper is well-written in general (Reviewer m5zL)
* the experiments support their results (Reviewer m5zL)
* the results in this paper seem quite interesting (Reviewer AQVy)
* it is nice to see more work remarking on the beneficial properties of importance sampling (Reviewer AQVy)
* the experimental evaluation shows the large difference between adversarially-robust and "non-robust" methods (Reviewer AQVy)

We provide our responses to the specific questions of each reviewer below. We hope our answers resolve all initial questions and concerns raised by the reviewers and we will be most happy to answer any remaining questions!

---

### Decision · Program_Chairs · 2021-09-27

**Decision:**

Accept (Poster)

**Comment:**

The submission provides a nice overall message that many streaming algorithms are already adversarially robust. This was deemed an important message. There were some concerns with the presentation, as discussed in the reviews and the follow-ups, which the authors are encouraged to address and correct.